# Cryo-EM structure of mammalian RNA polymerase II in complex with human RPAP2

Isaac Fianu [1], Christian Dienemann[1], Shintaro Aibara [1], Sandra Schilbach [1] & Patrick Cramer [1✉]

Nuclear import of RNA polymerase II (Pol II) involves the conserved factor RPAP2. Here we report the cryo-electron microscopy (cryo-EM) structure of mammalian Pol II in complex with human RPAP2 at 2.8 Å resolution. The structure shows that RPAP2 binds between the jaw domains of the polymerase subunits RPB1 and RPB5. RPAP2 is incompatible with binding of downstream DNA during transcription and is displaced upon formation of a transcription pre-initiation complex.

---

[1] Department of Molecular Biology, Max Planck Institute for Biophysical Chemistry, Göttingen, Germany. ✉email: patrick.cramer@mpibpc.mpg.de

RNA polymerase II (Pol II) is the enzyme that transcribes protein-coding genes in the nucleus to produce mRNA. A large amount of data has accumulated on Pol II structures that are relevant for understanding transcription[1]. However, the molecular basis of Pol II biogenesis and nuclear import remains poorly understood[2]. Currently, only one structure of Pol II in complex with a biogenesis factor is available, a low-resolution cryo-EM structure of yeast Pol II bound by the nuclear import factor Iwr1[3].

The RNA polymerase II associated protein 2 (RPAP2) was identified by co-purification with Pol II and its associated factors RPRD1A, RPRD1B, RPRD2, GRINL1A, and RECQL5[4,5]. RPAP2 shuttles between the cytoplasm and the nucleus and its silencing causes cytoplasmic accumulation of Pol II[6]. The RPAP2 homolog in *Arabidopsis thaliana* is also involved in nuclear import of Pol II[7] and its knockdown delays cell differentiation[8]. The yeast homolog of RPAP2 is called regulator of transcription 1 (Rtr1) and also shuttles between the cytoplasm and nucleus[9]. Rtr1 accumulates in the nucleus when the Pol II biogenesis factor Gpn1 is mutated[10] and the structure of Rtr1 is known[11–13].

RPAP2 and Rtr1 were also described as phosphatases that target the Pol II C-terminal repeat domain (CTD). For transcription of snRNA genes, RPAP2 may be recruited by serine-7 phosphorylation of the CTD, and dephosphorylates serine-5 residues[14,15]. Rtr1 from the yeast *Saccharomyces cerevisiae* was reported to have CTD phosphatase activity[16] and can dephosphorylate both tyrosine-1 and serine-5 residues in the CTD[12]. The structure of *S. cerevisiae* Rtr1 was reported to contain a phosphoryl transfer domain and an active site for an atypical phosphatase[13]. However, Rtr1 from the yeast *Kluyveromyces lactis* apparently lacks an active site and phosphatase activity[11]. The molecular basis of how RPAP2 (and its homologs) binds to Pol II to perform these functions is not known.

Here we report the cryo-EM structure of mammalian Pol II bound by human RPAP2. The structure shows that the highly conserved N-terminal domain of RPAP2 binds between the jaw domains of Pol II subunits RPB1 and RPB5 in the downstream region of the Pol II active center cleft. RPAP2 location on Pol II is incompatible with binding of the downstream DNA during transcription and it is displaced upon formation of a transcription pre-initiation complex (PIC).

## Results

### Cryo-EM analysis of Pol II–RPAP2 complex.
To investigate how RPAP2 interacts with Pol II, we aimed at localizing this factor on the Pol II surface. We prepared recombinant human RPAP2 after heterologous expression in insect cells and found that it readily bound to *Sus scrofa domesticus* Pol II that we purified from thymus samples as described[17]. A stable Pol II–RPAP2 complex could be purified by size-exclusion chromatography (Supplementary Fig. 1a). We prepared cryo-EM grids from purified Pol II–RPAP2 complex and obtained a single-particle cryo-EM reconstruction at a resolution of 2.8 Å from 365,000 particles (Fig. 1, Supplementary Fig. 1).

To build a model for the Pol II–RPAP2 complex, we first placed the structure of Pol II (PDB:5FLM)[17] into the density after nucleic acids in the model were removed. We then deleted the Pol II clamp and the RPB4/RPB7 stalk domains because density for these regions was absent or weak, respectively. After minor adjustments to the Pol II model, we placed a homology model for RPAP2 derived from the structure of yeast Rtr1[13] into the remaining density. We manually adjusted the homology model, guided by detailed density that revealed side chains (Supplementary Fig. 2). We observed unambiguous density for RPAP2 residues 48–155 and confidently modeled this region including

**Table 1 Cryo-EM data collection, refinement and validation statistics.**

| | Map 1 (EMD-12087, PDB 7B7U) | Map 2 (EMD-12087) |
|---|---|---|
| *Data collection and processing* | | |
| Magnification | 81,000 | |
| Voltage (kV) | 300 | |
| Electron exposure (e–/Å²) | 42.5 | |
| Defocus range (μm) | −0.5 to −3.2 | |
| Pixel size (Å) | 1.05 | |
| Symmetry imposed | C1 | |
| Initial particle images (no.) | 1,556,776 | |
| Final particle images (no.) | 364,771 | |
| Map resolution (Å) | 2.8 | 3.0 |
| FSC threshold | 0.143 | 0.143 |
| Map resolution range (Å) | 2.58–6.00 | 2.93–4.30 |
| Map sharpening *B* factor (Å²) | −55 | −113 |
| *Refinement* | | |
| Initial model (PDB) | 5FLM | |
| Model resolution (Å) | 3.1 | |
| Model composition | | |
| Non-hydrogen atoms | 25,142 | |
| Protein residues | 3,112 | |
| Ligands | 6 | |
| *B* factors (Å²) | | |
| Protein | 98.13 | |
| Ligand | 124.63 | |
| r.m.s. deviations | | |
| Bond lengths (Å) | 0.002 | |
| Bond angles (°) | 0.481 | |
| Validation | | |
| MolProbity score | 1.84 | |
| Clashscore | 8.72 | |
| Poor rotamers (%) | 0.07 | |
| Ramachandran plot | | |
| Favored (%) | 94.51 | |
| Allowed (%) | 5.49 | |
| Disallowed (%) | 0.0 | |

side chains. We extended the RPAP2 model by a few residues C-terminally using continuous density, but did not observe any density for the large C-terminal region of RPAP2 (Fig. 1a). The structure was real-space refined and has good stereochemistry (Table 1).

### Structure of Pol II–RPAP2 complex.
The structure shows that RPAP2 occupies a large part of the downstream region of the active center cleft of Pol II (Fig. 1b). The conserved N-terminal region of RPAP2 binds between the jaw domains of the Pol II subunits RPB1 and RPB5. This is consistent with the previous finding that it is the N-terminal region of RPAP2 that mediates Pol II binding[6]. This N-terminal region of RPAP2 consists of four helices that are conserved in yeast Rtr1[11–13]. In addition, the N-terminal region of RPAP2 contains an 'insertion' (residues 116–128) that forms two β-strands and thus deviates from the corresponding region in *S. cerevisiae* Rtr1, which forms an α-helix[13] (Fig. 1b and Supplementary Fig. 3a). Residues in the RPAP2 insertion are conserved from *Drosophila* to human (Supplementary Fig. 3b), but not in yeast, suggesting it is a metazoan-specific feature of RPAP2. The region of RPAP2 following C-terminally of the conserved N-terminal domain extends to the opposite side of the cleft to the RPB1 jaw-lobe module (Fig. 1b).

Our structure also reveals details of the interaction of RPAP2 with Pol II. The RPAP2 insertion contacts helix α43 in the RPB1

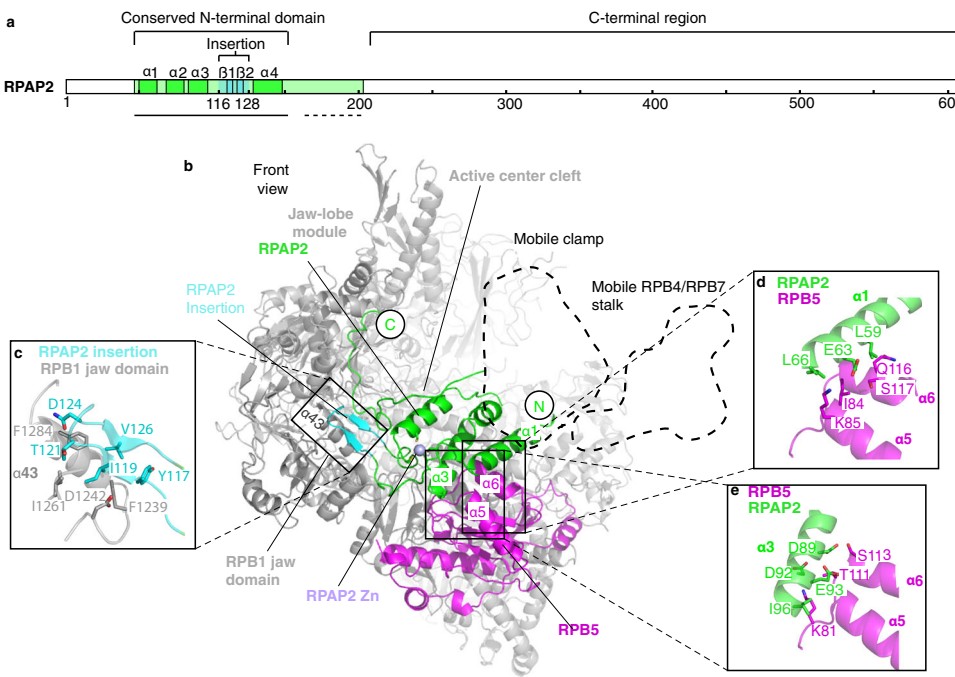

**Fig. 1 Structure of Pol II–RPAP2 complex. a** Primary architecture of RPAP2. The color code is used throughout. The black line below indicates confidently modeled regions. Dashed black line indicates region with backbone trace. α-helices and the insertion β-strands are indicated. **b** Ribbon model viewed from the front (Pol II, gray except RPB5, magenta; RPAP2, green except RPAP2 insertion, cyan). The RPAP2 zinc metal is displayed as a light blue sphere. The canonical location of the Pol II mobile clamp and the RPB4/RPB7 stalk are outlined by dashed lines. This and other structural figure panels were prepared with PyMol (version 2.3.4). **c–e** Detailed views of interactions between RPAP2 insertion and RPB1 jaw domain and between RPAP2 and RPB5. Side chains of residues in the interaction interfaces are shown as sticks and are labeled.

jaw domain and buries a surface area[18] of ~580 Å² (Fig. 1c). The contacts between this metazoan-specific region of RPAP2 and RPB1 may lead to a stronger binding of RPAP2 to Pol II in metazoans compared to binding of Rtr1 to yeast Pol II. The interaction of RPAP2 with RPB5 is mediated by helices α1 and α3 of RPAP2 and helices α5 and α6 of RPB5 (Fig. 1d and e). The interface between RPAP2 and RPB5 includes hydrophobic, ionic, and hydrogen bonding interactions, burying a total surface area[18] of ~520 Å². RPAP2 residues L59, L66, and I96 are in van der Waals contact with parts of RPB5 residues Q116, I84, and K81, respectively. RPAP2 residues E63 and D92 form salt bridges with RPB5 residues K85 and K81, respectively. In addition, RPAP2 residues D89 and E93 form hydrogen bonds with RPB5 residues S113 and T111, respectively. Mutation of residue E66 to alanine in yeast Rtr1 (corresponding to E93 in RPAP2) led to a growth defect phenotype similar to Rtr1 deletion[12] demonstrating the importance of this interaction. The interacting residues are conserved across species (Supplementary Fig. 3b), indicating that the contacts observed here are highly conserved, and pointing to an evolutionary conserved function of RPAP2.

**RPAP2 is displaced by downstream DNA upon transcription.** The location of RPAP2 between the jaws of the Pol II active center cleft suggested that RPAP2 interferes with binding of downstream DNA during transcription. Indeed, superposition of our Pol II-RPAP2 structure with the structure of the Pol II elongation complex (EC)[19] revealed severe clashes between RPAP2 and downstream DNA (Fig. 2a). We therefore tested whether a DNA–RNA scaffold that mimics the nucleic acids during transcription elongation would displace RPAP2 from Pol II. We prepared a RPAP2 variant, RPAP2(1–215), which includes the region of RPAP2 observed in our structure, and found that this variant readily bound Pol II (Fig. 2c, Peak 4). Incubation of a Pol II–RPAP2(1–215) complex with the DNA–RNA scaffold

displaced RPAP2(1–215) from Pol II (Fig. 2c, Peak 1). These results indicate that RPAP2 binding to Pol II is incompatible with formation of a Pol II EC and transcription.

We next asked whether RPAP2 may already be displaced from Pol II upon formation of a transcription PIC. Consistent with this idea, superposition of our Pol II-RPAP2 structure onto the structure of the Pol II core PIC[20] showed strong clashes of RPAP2 with downstream promoter DNA (Fig. 2b). To test whether core PIC formation displaces RPAP2 from Pol II, we incubated the Pol II–RPAP2(1–215) complex with the transcription initiation factor (TF)-IIF and preformed upstream promoter complex containing promoter DNA, TFIIA, TFIIB, and the TATA box-binding protein (TBP). This led to displacement of RPAP2(1–215) from Pol II and formation of a core PIC (Fig. 2c, Peak 2). In contrast, double-stranded promoter DNA alone could not displace RPAP2 (1–215) from Pol II (Fig. 2c, Peak 3). Together, these results show that PIC formation displaces RPAP2 from Pol II before transcription starts. The results are consistent with the published observation that RPAP2 does not copurify with the Mediator complex[5], which is known to bind the PIC.

**Discussion**
In conclusion, our work provides the structure of human RPAP2 and unambiguously localizes it on the Pol II surface between the jaw domains of RPB1 and RPB5. The RPAP2 location is incompatible with the presence of downstream DNA during transcription and is consistent with RPAP2 being a factor involved in Pol II biogenesis and nuclear import. In addition, our results suggest that formation of the PIC liberates RPAP2 for its recycling by nuclear export. In summary, our findings and published data converge on the following model. RPAP2 enters the nucleus in association with Pol II, and its conserved N-terminal domain is displaced upon assembly of the PIC for transcription initiation. After dissociation from Pol II, RPAP2 returns to the

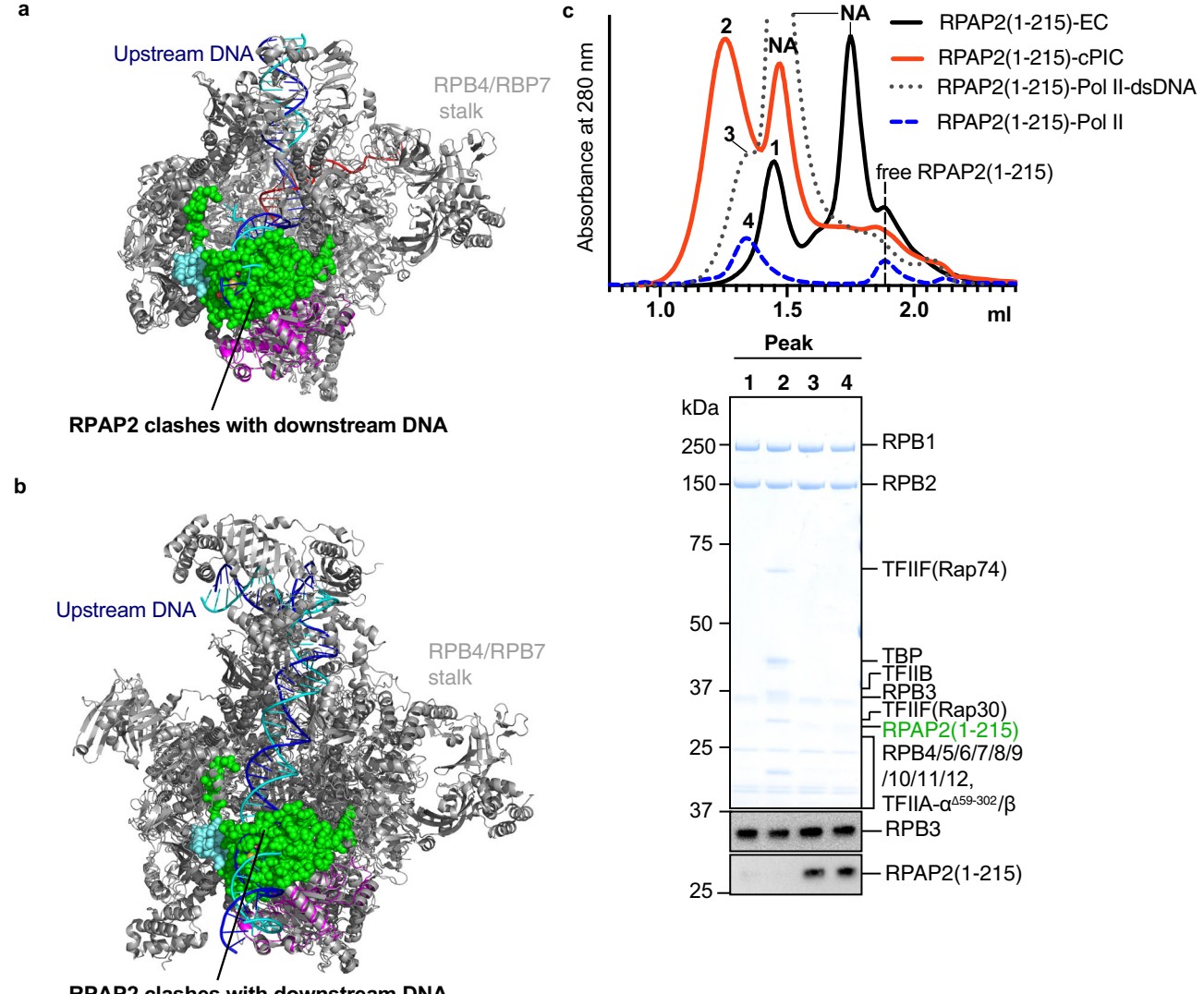

**Fig. 2 RPAP2 binding is incompatible with transcription initiation and elongation. a** Superposition of the Pol II-RPAP2 structure onto the Pol II elongation complex (PDB: 5FLM) reveals that RPAP2 would clash with downstream DNA (blue/cyan). **b** Superposition of the Pol II-RPAP2 structure onto the Pol II pre-initiation complex (PDB: 5IYA) reveals that RPAP2 would clash with downstream DNA (blue/cyan). **c** Binding competition assays using the preformed Pol II–RPAP2(1–215) complex (peak 4) show that RPAP2 is displaced from Pol II upon formation of an elongation complex (peak 1) or a pre-initiation complex (peak 2) but cannot be displaced by double-stranded promoter DNA (peak 3). Chromatograms show the formation of complexes and the relevant peak fractions used for SDS–PAGE and western blot analysis are indicated. Peaks for free nucleic acids are indicated by NA and vertical dashed lines show the elution peak of free RPAP2(1-215). A representative western blot analysis of the same peak fractions using anti RPAP2 (Thermo Fisher #PA5-61244) and anti RPB3 (BETHYL #A303-771A) antibodies are shown. Please refer to Supplementary Fig. 5 for source data of western blots.

cytoplasm in a manner that depends on the GTPase GPN1[6] and remains to be studied.

Our structure of the Pol II–RPAP2 complex and complementary biochemical data thus is consistent with RPAP2 functioning primarily as a Pol II biogenesis factor that is displaced when Pol II engaged with DNA for transcription. It is however not excluded that RPAP2 remains bound to a transcription EC under certain circumstances because the C-terminal region of RPAP2, which is not involved in the interactions observed here structurally, may make additional contacts with Pol II[21]. Furthermore, RPAP2 may indirectly interact with hyperphosphorylated Pol II via the proteins RPRD1A and RPRD1B[5,15], allowing it to dephosphorylate the Pol II CTD. Unfortunately, our structure does not provide new insights into the phosphorylation-related functions of RPAP2.

## Methods

**Molecular cloning and protein expression**. A synthetic DNA fragment encoding the full-length *Homo sapiens* RPAP2 codon optimized for insect cell expression was purchased from Integrated DNA Technologies (IDT). The DNA fragment was inserted into insect cell expression vector 438-C (Addgene: Plasmid #55220) by ligation independent cloning (LIC)[22]. Similarly, a DNA fragment encoding RPAP2 (1–215) codon optimized for *Escherichia coli* expression was purchased from IDT and cloned into vector 1-B (Addgene: Plasmid #29653) by LIC. Both fragments were designed to contain LIC compatible overhangs for cloning into the stated vectors. *H. sapiens* TFIIA-α variant (TFIIA-α$^{Δ59-302}$) in which the unstructured insertion (residues 59–302) was replaced by a linker (1–58—LEVLFQGP—303-376-6xHis) was cloned in the first multiple cloning site (MCS1) of pET-Duet vector and full-length TFIIA-β-6xHis in MCS2. Both genes were codon-optimized for *E. coli* expression. All clones were verified by Sanger sequencing.

Full-length RPAP2 was expressed in insect cells. Production of bacmid, $V_0$ and $V_1$ baculoviruses, and protein expression were performed as previously described[23]. Hi5 cells were harvested by centrifugation, cells were suspended in a lysis buffer (35 mL/L of culture) containing 25 mM Tris–HCl pH 8, 500 mM NaCl, 30 mM

imidazole pH 8, 10% glycerol, 10 mM beta-mercaptoethanol (BME), 10 µM ZnCl$_2$, 0.284 µg/mL leupeptin, 1.37 µg/mL pepstatin A, 0.17 mg/mL PMSF, and 0.33 mg/mL benzamidine), flash-frozen in liquid nitrogen, and stored at −80 °C until purification. RPAP2(1–215) was expressed in *E. coli* as described[11] without addition of 0.1 mM ZnCl$_2$ in the expression medium.

The TFIIA variant was expressed in BL21 *E. coli* by induction at OD600 of 0.8 with 500 µM IPTG for 18 h at 18 °C. Harvested cells were resuspended in buffer H$_E$-30 (25 mM HEPES pH 7.6, 500 mM KCl, 10% glycerol (v/v), 30 mM imidazole, 5 mM BME, 0.284 µg/mL leupeptin, 1.37 µg/mL pepstatin A, 0.17 mg/mL PMSF, and 0.33 mg/mL benzamidine), and stored at −80 °C until purification.

Cloning and expression of human TBP, TFIIB, and TFIIF will be described elsewhere.

**Protein purification.** All protein purification steps were performed at 4 °C unless stated otherwise.

Frozen pellets of Hi5 cells expressing full-length RPAP2 were thawed at room temperature and lysed by sonication. The lysate was cleared by centrifugation. Clarified lysate was filtered through 0.8 µm syringe filters (Merck Millipore) and applied to a 5 mL HisTrap HP column (GE Healthcare) equilibrated in lysis buffer. The column was washed with 50 mL of wash buffer containing 25 mM Tris–HCl pH 8, 500 mM NaCl, 30 mM imidazole pH 8, 10% glycerol (v/v), 10 mM BME, 10 µM ZnCl$_2$). A self-packed amylose column with a total bed volume of 15 mL amylose beads (NEB) was equilibrated in wash buffer and connected in tandem to the 5 mL HisTrap column. Bound protein was then eluted from the 5 mL HisTrap column using wash buffer supplemented with 250 mM imidazole. The HisTrap column was detached and the amylose column was washed with 100 mL of wash buffer before eluting with wash buffer supplemented with 100 mM maltose. Peak fractions were analyzed by SDS–PAGE and Coomassie staining. Fractions containing RPAP2 were pooled and treated overnight with 6xHis tagged TEV protease. The protein was then applied to a 5 mL HisTrap column equilibrated in wash buffer to remove the tags, TEV protease and protein with uncleaved tags. The flow through was collected and concentrated using a 30,000 MWCO Amicon Ultra Centrifugal Filter (Merck Millipore). The protein was applied to a Superdex 200 10/300 increase column (GE Healthcare) equilibrated in size exclusion buffer (20 mM HEPES pH 7.4, 200 mM NaCl, 10% (v/v) glycerol, 10 mM BME, 10 µM ZnCl$_2$). Peak fractions were assessed by SDS–PAGE and Coomassie staining. Peak fractions containing the protein of interest were pooled and concentrated using a 30,000 MWCO Amicon Ultra Centrifugal Filter (Merck Millipore), aliquoted, snap frozen in liquid nitrogen, and stored at −80 °C.

We purified RPAP2(1–215) using the same protocol as for the full-length RPAP2 but omitting the amylose affinity step.

Mammalian Pol II was purified from *S. scrofa* thymus essentially as described[17]. Gdown1 containing fractions were discarded during the Uno Q ion exchange step. Peak fractions containing Pol II were concentrated using a 100 K MWCO Amicon Ultra Centrifugal Filter (Merck Millipore) and buffer exchanged into Pol II dilution buffer (10 mM HEPES pH 7.25, 150 mM NaCl, 10% glycerol, 10 µM ZnCl$_2$, and 1 mM DTT). Protein was aliquoted, flash-frozen in liquid nitrogen, and stored at −80 °C.

Frozen pellets of *E. coli* expressing the TFIIA variant were thawed at room temperature and lysed by sonication. The cell lysate was cleared by centrifugation. The soluble fraction was filtered through 0.45 µm syringe filters (Merck Millipore) and applied to a 5 mL HisTrap HP pre-equilibrated in buffer H$_E$-30. After sample application the column was washed with 50 mL of buffer H$_E$-30, 25 mL of buffer H$_E$-30S (25 mM HEPES pH 7.6, 1000 mM KCl, 10% glycerol (v/v), 30 mM imidazole, 5 mM BME, 0.284 µg/mL leupeptin, 1.37 µg/mL pepstatin A, 0.17 mg/mL PMSF and 0.33 mg/mL benzamidine) and 25 mL of buffer HEP$_E$-100 (25 mM HEPES pH 7.6, 100 mM KCl, 5% glycerol (v/v), 5 mM BME). Protein was eluted with a linear gradient of 0–100% buffer H$_E$-500 (25 mM HEPES pH 7.6, 100 mM KCl, 5% glycerol (v/v), 500 mM imidazole, 5 mM BME) over 60 mL. The eluted sample was applied to a 1 mL HiTrap Heparin HP column (GE Healthcare) pre-equilibrated in buffer HEP$_E$-100. The column was washed with 50 mL of buffer HEP$_E$-100 and the protein was eluted with a linear gradient of 0–40% buffer HEP$_E$-1000 (25 mM HEPES pH 7.6, 1000 mM KCl, 5% glycerol (v/v), 5 mM BME) over 30 mL. Fractions containing both TFIIA subunits were pooled, diluted to reach a salt concentration of 100 mM KCl and applied to a 1 mL HiTrap SP HP column (GE Healthcare) pre-equilibrated in buffer HEP$_E$-100. The column was washed with 10 mL of buffer HEP$_E$-100 and the protein was eluted with a linear gradient of 0–70% buffer HEP$_E$-1000 over 70 mL. Fractions containing stoichiometric TFIIA variant were pooled, concentrated with a 10,000 MWCO Amicon Ultra Centrifugal Filter (Merck Millipore) and applied to a Superdex75 10/300 GL size exclusion column (GE Healthcare) pre-equilibrated in size exclusion buffer (25 mM HEPES, pH 7.6, 200 mM KCl, 5% glycerol (v/v), 2 mM TCEP). Peak fractions were pooled, concentrated with a 10,000 MWCO Amicon Ultra Centrifugal Filter (Merck Millipore), aliquoted, flash-frozen in liquid nitrogen and stored at −80 °C. Purification of human TBP, TFIIB, and TFIIF will be described elsewhere.

Representative SDS–PAGE analyses of the purified proteins are shown in Supplementary Fig. 4.

**Analytical size-exclusion chromatography and competition assays.** To test binding of RPAP2 and RPAP2(1–215) to Pol II, we mixed 30 pmol Pol II with 150 pmol of RPAP2/RPAP2(1–215), diluted to a final volume of 40 µL (final conditions: 20 mM HEPES pH 7.4, 100 mM NaCl, 4% (v/v) glycerol, 3 mM MgCl$_2$, and 1 mM TCEP) and incubated at 30 °C for 30 min. Samples were applied to a Superose 6 increase 3.2/300 column (GE Healthcare) equilibrated in complex buffer (20 mM HEPES pH 7.4, 100 mM NaCl, 4 % (v/v) glycerol, 3 mM MgCl$_2$, and 1 mM TCEP) at 4 °C. Peak fractions were analyzed by SDS–PAGE followed by Coomassie staining.

For competition assays between RPAP2(1–215) and double-stranded DNA (dsDNA), we used oligos of the adenovirus late promoter (purchased from IDT) with the following sequences; template:

5′AGGGAGTACTCACCCCAACAGCTGGCCCTCGCAGACAGCGATGCGG
AAGAGAGTGAGGACGAACGCGCCCCCACCCCCTTTTATAGCCCCCCTTC
AGGAACACCCG 3′; non-template:

5′CGGGTGTTCCTGAAGGGGGGCTATAAAAGGGGGTGGGG
GCGCGTTCGTCCTCACTCTCTTCCGCATCGCTGTCTGCGAGGGC-
CAGCTGTTGGGGTGAGTACTCCCT 3′. We annealed the template and non-template to a final concentration of 50 µM. We mixed Pol II (30 pmol), RPAP2 (1–215) and dsDNA in a 1:5:5 molar ratio. The reaction conditions were as described for Pol II–RPAP2 binding. After 30 min incubation at 30 °C, the sample was applied to a Superose 6 increase 3.2/300 column equilibrated in complex buffer at 4 °C. Peak fractions were analyzed by SDS–PAGE followed by Coomassie staining.

To test binding of RPAP2(1–215) to Pol II upon formation of a core PIC, we first tested the formation of a Pol II–TFIIF–RPAP2 ternary complex. We mixed Pol II, RPAP2(1–215), and TFIIF and incubated for 30 min at room temperature (RT). We could not form a stable Pol II–TFIIF–RPAP2 ternary complex. In parallel, dsDNA (same as above), TFIIA variant, TFIIB and TBP were mixed and incubated for 2 min at RT to form an upstream promoter complex. We then mixed the Pol II–RPAP2(1–215) complex, TFIIF and the upstream promoter complex, diluted to a final volume of 40 µL in complex buffer and incubated for 30 min at 30 °C. The complex was applied to a Superose 6 increase 3.2/300 and analyzed as described above. All factors including dsDNA were added in 5 molar excess relative to Pol II.

To test binding of RPAP2(1–215) to the Pol II EC, EC was formed as described with 5 molar excess of pre-annealed template DNA–RNA and non-template DNA[24] in the presence of 5 molar excess of RPAP2(1–215). Nucleic acid scaffolds (purchased from IDT) with the following sequences were used. template DNA: 5′ GCTCCCAGCTCCCTGCTGGCTCCGAGTGGGTTCTGCCGCTCTCAATGG 3′; RNA: 5′ GAAUAACCGGAGAGGGAACCCACU 3′; non-template DNA: 5′ CCATTGAGAGCGGCCCTTGTGTTCAGGAGCCAGCAGGGAGCTGGGAGC 3′. Binding conditions were as described above for the Pol II–RPAP2 complex. Peak fractions were analyzed by SDS–PAGE and Coomassie staining.

Peak fractions from each competition assay were analyzed by SDS–PAGE and western blotting. One SDS–PAGE was run for each western blot. Protein bands from SDS–PAGE were transferred to nitrocellulose membranes (one for each gel) using the Trans-Blot Turbo Transfer System™ (Bio-Rad). Membranes were blocked with 2% milk in PBS-T (1× PBS and 0.1% Tween20) on a shaker at room temperature for 1 h. Rabbit anti RPAP2 (Thermo Fisher Scientific #PA5-61244) and rabbit anti-RPB3 (BETHYL #A303-771A) antibodies were added to the blocked membranes (an antibody/membrane) and incubated at 4 °C overnight. Unbound primary antibodies were discarded and the membranes were washed 3× with PBS-T. The membranes were then incubated with donkey anti-rabbit secondary antibody conjugated to horseradish peroxidase (HRP) at room temperature for 2 h. Free secondary antibody was discarded and the membranes were washed 3× with PBS-T. The blots were developed with 1 mL each of peroxide and enhancer solution (Thermo Fisher Scientific) and imaged for chemiluminescence using an Intas imager.

**Sample preparation for cryo-EM.** Formation of Pol II–RPAP2 complex for cryo-EM analysis was done as described in the binding assay but with 128 pmol Pol II and 640.5 pmol of RPAP2. The peak fraction containing the complex was cross-linked with 1 mM BS3 (bis(sulfosuccinimidyl)suberate) (Thermo Fisher Scientific) on ice for 30 min and quenched with 50 mM ammonium bicarbonate. The crosslinked sample was dialyzed for 5 h at 4 °C against the complex buffer without glycerol using 20,000 MWCO Slide-A-Lyzer MINI Dialysis Unit (Thermo Fisher Scientific). 2 µL of dialyzed sample was applied to each side of freshly glow-discharged R2/2 UltrAuFoil grids (Quantifoil). Grids were blotted for 6 s with a blot force of 5 before vitrified by plunging into liquid ethane using a Vitrobot Mark IV (FEI Company) operated at 4 °C and 100% humidity.

**Cryo-EM data collection and processing.** Cryo-EM data collection was performed with SerialEM[25] using a Titan Krios transmission electron microscope (Thermo Fisher Scientific) operated at 300 kV. Images were acquired in EFTEM mode with a slit width of 20 eV using a Quantum LS energy filter and a K3 direct electron detector (Gatan) at a nominal magnification of ×81,000 corresponding to a calibrated pixel size of 1.05 Å/pixel. Exposures were recorded in counting mode for 2.21 s with a dose rate of 21.05 e−/pixel/s resulting in a total dose of 42.4 e−/Å$^2$ which was fractionated into 40 movie frames. Motion correction, dose weighting, CTF estimation, and particle picking were performed on-the-fly using Warp[26]. About 100,000 particles were initially extracted in Warp using a box size of 360 pixels. This particle set was subjected to 2D classification in cryoSPARC[27] and an

ab initio 3D model was generated from particles selected from good 2D classes. ~1.6 million particles were extracted in Relion 3.0[28] in a box of 180 pixels and binned to a pixel size of 2.1 Å/pixel. Extracted particles were subjected to 3D classification using a 60 Å low-pass filtered ab initio model generated in cryoS-PARC as reference to eliminate bad particles. For further cleaning of the data, selected good particles were subjected to 2D classification in Relion limiting the resolution to 10 Å. The resulting 1.2 million good particles were re-extracted without binning and refined using the ab initio map scaled and low-pass filtered to 60 Å as a reference. To improve the resolution of density corresponding to RPAP2, we applied a soft mask encompassing RPB5, RPAP2 density, and the RPB1 jaw-lobe module (Supplementary Fig. 1f) and performed 3D classification without image alignment. We performed a 3D refinement of the best class containing 365,000 particles to obtain a 2.8 Å reconstruction (map 1). We then performed a focused 3D refinement with mask to obtain a 3.0 Å reconstruction of the region around RPB5, RPAP2, and the RPB1 jaw-lobe module (map 2). Density corresponding to the Pol II clamp and RPB4/RPB7 stalk domains were very noisy and weak suggesting mobility and/or flexibility of these domains. We therefore excluded them in our modeling.

**Model building and refinement**. To build an atomic model of the Pol II–RPAP2 complex, we used the structure of transcribing Pol II (PDB: 5FLM)[17] after nucleic acids, Pol II clamp and RPB4/RPB7 stalk domains were deleted as an input model. The structure was rigid-body docked into the global cryo-EM reconstruction (map 1) in Chimera[29] and manually adjusted in Coot[30] to improve the density-fit. Non-conserved residues between bovine and porcine Pol II were mutated to porcine residues. The model was real-space refined in PHENIX[31]. To obtain a model for RPAP2, we generated a homology model based on the crystal structure of Rtr1 (PDB:5C2Y)[13] using SWISS-MODEL[32]. The homology model was rigid body docked into the focus cryo-EM map (map 2) in Chimera and manually adjusted in Coot. Regions corresponding to the insertion (residues 116–128) which deviated substantially from the homology model were built de-novo into the density. Since we could not unambiguously assign the sequence register based on the density beyond residue 155, we traced the protein backbone in the density for the remaining parts of the RPAP2 model. To optimize the geometry, we real-space refined the model against map 2 in PHENIX. Models for Pol II and RPAP2 were merged into one molecule (in map 1) using Coot and subjected to a final round of PHENIX real-space refinement against the global map (map 1) using standard parameters and protein secondary structure restraints.

**Reporting summary**. Further information on experimental design is available in the Nature Research Reporting Summary linked to this article.

## Data availability
The cryo-EM reconstructions and final model were deposited with the Electron Microscopy Data Base (EMDB) under accession code EMD-12087 and with the Protein Data Bank (PDB) accession 7B7U. Source data for Fig. 2c are available with the paper online.

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

## Acknowledgements
We thank F. Grabbe for purifying initiation factors, U. Neef and P. Rus for maintaining insect cell facility, T. Schulz for maintaining cells and pig thymus stocks, C. Dienemann and U. Steuerwald for maintaining the EM facility. We thank Gaurika Garg and Srini-vasan Rengachari for nucleic acids used in EC and cPIC formation respectively. We thank members of the Cramer lab for general discussion of the data. P.C. was supported by the Deutsche Forschungsgemeinschaft (SFB860, SPP2191, EXC 2067/1-390729940) and the ERC Advanced Investigator Grant CHROMATRANS (grant agreement No 693023). S.A. was supported by H2020 Marie Curie Individual Fellowship (894862).

## Author contributions
I.F. conceived, designed, and executed all experiments and analyzed data, unless stated otherwise. I.F. and C.D. collected cryo-EM data. I.F. built model with help from S.A. and S.S. S.S. established human initiation factors preparations and purified TFIIA-α$^{\Delta59-302}$. P.C. supervised research. I.F. and P.C. interpreted the data and wrote the manuscript with input from all authors.

## Funding

## Competing interests

The authors declare no competing interests.
