## [Peer Review File · Communications Biology]

Reviewers' comments:

Reviewer #1 (Remarks to the Author):

The manuscript by Fianu and colleagues represents the first high resolution structure of the metazoan RNA polymerase II (Pol II) in complex with a biogenesis factor. The major finding of the manuscript is that the binding of this biogenesis factor, RPAP2, to Pol II is impeding the binding of downstream DNA to the Pol II active center cleft.

The structural data are of high quality.

I have the following comments:

- 1) RPAP2 was also described as a Pol II CTD phosphatase, potentially removing the phosphate group of the Ser5-P during elongation. The authors show that RPAP2 is displaced during elongation, seemingly contradicting this finding. However, the manuscript fails to discuss that RPAP2 is a potential CTD phosphatase during elongation.
- 2) The authors don't detail why in their binding experiments they used a truncated protein, only about one third of the RPAP2. The yeast homologue of RPAP2, Rtr1, was reported to be able to bind to hyperphosphorylated RNAPII. However, RPAP2 was reported to interact with the phosphorylated CTD through two other proteins, RPRD1A/B (Ni et al. Nat. Struct. Mol. Biol., 21 (2014), 686-695). In my view, the presented binding experiment with the truncated RPAP2 doesn't exclude the possibility that the full-length RPAP2 is interacting with the hyperphosphorylated CTD of the elongation complex through RPRD1A/B. The authors should discuss this.
- 3) In my opinion it is not easy to make the connection between the overall structure, depicted on Figure 1/b and the image showing the interaction details (Figure 1/e and f).

Reviewer #2 (Remarks to the Author):

In this manuscript entitled 'Structure of RNA polymerase II-RPAP2 complex', by Fianu et al., the authors report the cryo-EM structure of the conserved N-terminal domain of human RPAP2 in complex with the pig RNA polymerase II (Pol II). Their model clearly shows that a conserved N-terminal domain of RPAP2 binds to the jaw domains of RPB1 and RPB5, thereby preventing insertion of DNA in the Pol II cleft. Indeed, competition experiments revealed that RPAP2 dissociated from Pol II during preinitiation complex (PIC) formation. Although the crystal structure of the N-terminal domain of the yeast homolog of RPAP2 (Rtr1) has been previously reported (ref 11-13 in this manuscript), the present study convincingly visualizes the interaction between the RPAP2 and Pol II. These results suggest a model in which RPAP2 acts as a Pol II biogenesis factor involved in Pol II nuclear import and then dissociates from Pol II upon PIC formation. The manuscript is well written and will be of interest to researchers in the transcription field.

However, there are some important questions regarding the role of RPAP2 which are curiously overlooked.

1. There is an abundant literature indicating that RPAP2 and its yeast homolog Rtr1 are phosphatases acting on Pol II CTD, more specifically on Serine 5 phosphorylated CTD (Mosley et al. 2010, PMID 19394294; Egloff et al. 2012, PMID: 22137580, Ni et al 2014, PMID: 24997600 and ref 12 and 13 in this manuscript). Although a crystal structure of *Kluyveromyces lactis* Rtr1 suggested that it lacks phosphatase activity, a more recent study on the *S. cerevisiae* protein confirmed the presence of an active site capable of a phosphoryl transfer reaction (ref 11 and 13 of this manuscript). Could the authors comment on the putative phosphatase activity of RPAP2 in light of their data? Does their structure allow to reject or support an enzymatic activity of RPAP2? Could RPAP2 have two independent functions related to different stages of Pol II transcription cycle?
2. Hsu et al. (2014; ref 12 in this manuscript) reported that a point mutation in the yeast RTR1 gene (Rtr1-E66A) causes a growth defect similar to that of RTR1 deletion (*rtr1Δ*). This residue is conserved in RPAP2 (E93) and was found in this study to interact with RPB5 (see Extended Data Figure 3). Can

the authors specify if mutation of this residue would affect RPAP2 interaction with Pol II? In relation to this, does the binding of RPAP2 to the Pol II jaw require phosphorylation of the Pol II subunits, e.g. phosphorylation of T111 of RPB5 (situated opposite to E93)? More specifically, could the authors observe any density in their cryo-EM structure that could correspond to phosphorylation of e.g. T111 of RPB5?

3. It is unclear to me, whether the competition assays address the association of RPAP2-Pol II with TFIIF. Methods and Results of competition assays could be better explained. Could the authors clarify if binding of TFIIF to Pol II would be sufficient for the dissociation of RPAP2 from Pol II (see line 71-73)? More specifically, could Pol II be simultaneously bound by RPAP2 and TFIIF. As it was reported that a significant fraction of free Pol II in the nucleus associates with TFIIF, it should be possible to assess whether a ternary Pol II-TFIIF-RPAP2 can be purified (Rani et al. 2004, PMID: 14749386).

4. Could the authors discuss if the structural differences of the insertion between RPAP2 (β -sheets) and Rtr1 (α -helix) would result into functional differences (stronger/weaker interactions with RPB1 etc.)?

Response to reviewer comments
Manuscript COMMSBIO-20-3505-T
Fianu et al. “Structure of RNA polymerase II-RPAP2 complex”

Responses are in italics

Reviewer #1 (Remarks to the Author):

The manuscript by Fianu and colleagues represents the first high resolution structure of the metazoan RNA polymerase II (Pol II) in complex with a biogenesis factor. The major finding of the manuscript is that the binding of this biogenesis factor, RPAP2, to Pol II is impeding the binding of downstream DNA to the Pol II active center cleft.

The structural data are of high quality.

I have the following comments:

1) RPAP2 was also described as a Pol II CTD phosphatase, potentially removing the phosphate group of the Ser5-P during elongation. The authors show that RPAP2 is displaced during elongation, seemingly contradicting this finding. However, the manuscript fails to discuss that RPAP2 is a potential CTD phosphatase during elongation.

We now included the following sentences and further discussion in response to comment 2: Line 24: “RPAP2 and Rtr1 were also described as non-canonical phosphatases that target the Pol II C-terminal repeat domain (CTD). During transcription of snRNA genes, RPAP2 may be recruited by serine-7 phosphorylation of the CTD, and dephosphorylates serine-5 residues (PMID: 22137580, PMID 24997600). Rtr1 from the yeast Saccharomyces cerevisiae was reported to have CTD phosphatase activity (19394294) and can dephosphorylate both tyrosine-1 and serine-5 residues in the CTD (24951832). The structure of Saccharomyces cerevisiae Rtr1 identified a phosphoryl transfer domain and revealed an active site for an atypical phosphatase that may be conserved in RPAP2 (PMID:26933063). However, Rtr1 from the yeast Kluyveromyces lactis apparently lacks an active site and phosphatase activity (22781759). Unfortunately, our structure does not provide new insights into this function of the protein”

2) The authors don't detail why in their binding experiments they used a truncated protein, only about one third of the RPAP2. The yeast homologue of RPAP2, Rtr1, was reported to be able to bind to hyperphosphorylated RNAPII. However, RPAP2 was reported to interact with the phosphorylated CTD through two other proteins, RPRD1A/B (Ni et al. Nat. Struct. Mol. Biol., 21 (2014), 686-695). In my view, the presented binding experiment with the truncated RPAP2 doesn't exclude the possibility that the full-length RPAP2 is interacting with the hyperphosphorylated CTD of the elongation complex through RPRD1A/B. The authors should discuss this.

We used the truncated protein RPAP(1-215) in order to relate our functional to our structural data – it is this region of the protein that we observe to be ordered in our structure. Our data shows that this conserved N-terminal region must be displaced during transcription due to a clash with downstream DNA. With respect to the reviewer's other point, we think RPAP2 may tether the polymerase via its C-terminal region or via other proteins such as RPRD1A and RPRD1B during transcription.

We now included the following in the manuscript:

Line 94: It is however not excluded that RPAP2 remains bound to an elongation complex. The C-terminal region of RPAP2 may make additional contacts with Pol II [PMID: 25639305] which may allow RPAP2 to bind to an elongation complex. Furthermore, When RPAP2 is displaced by downstream DNA, RPAP2 may indirectly interact with hyperphosphorylated Pol II via RPRD1A and RPRD1B [PMID: 22231121, PMID: 24997600] allowing it to dephosphorylate the Pol II CTD.

3) In my opinion it is not easy to make the connection between the overall structure, depicted on Figure 1/b and the image showing the interaction details (Figure 1/e and f).

We modified the Figure for clarity: Figure 1c in now moved to Extended Data Figure 3a. We also provided rectangles on Figure 1b that correspond to Figure 1c, d and e respectively that illustrate interaction details.

Extended Data Figure 3

Reviewer #2 (Remarks to the Author):

In this manuscript entitled 'Structure of RNA polymerase II-RPAP2 complex', by Fianu et al., the authors report the cryo-EM structure of the conserved N-terminal domain of human RPAP2 in complex with the pig RNA polymerase II (Pol II). Their model clearly shows that a conserved N-terminal domain of RPAP2 binds to the jaw domains of RPB1 and RPB5, thereby preventing insertion of DNA in the Pol II cleft. Indeed, competition experiments revealed that RPAP2 dissociated from Pol II during preinitiation complex (PIC) formation. Although the crystal structure of the N-terminal domain of the yeast homolog of RPAP2 (Rtr1) has been previously reported (ref 11-13 in this manuscript), the present study convincingly visualizes the interaction between the RPAP2 and Pol II. These results suggest a model in which RPAP2 acts as a Pol II biogenesis factor involved in Pol II nuclear import and then dissociates from Pol II upon PIC formation. The manuscript is well written and will be of interest to researchers in the transcription field.

However, there are some important questions regarding the role of RPAP2 which are curiously overlooked.

1. There is an abundant literature indicating that RPAP2 and its yeast homolog Rtr1 are phosphatases acting on Pol II CTD, more specifically on Serine 5 phosphorylated CTD (Mosley et al. 2010, PMID 19394294; Egloff et al. 2012, PMID: 22137580, Ni et al 2014, PMID: 24997600 and ref 12 and 13 in this manuscript). Although a crystal structure of *Kluyveromyces lactis* Rtr1 suggested that it lacks phosphatase activity, a more recent study on the *S. cerevisiae* protein confirmed the presence of an active site capable of a phosphoryl transfer reaction (ref 11 and 13 of this manuscript).

Could the authors comment on the putative phosphatase activity of RPAP2 in light of their data?

Does their structure allow to reject or support an enzymatic activity of RPAP2?

Could RPAP2 have two independent functions related to different stages of Pol II transcription cycle?

We now discuss these important points in the text. Please compare our answers to reviewer #1. Briefly, we do not have insights into a phosphatase function of the protein, but we now mention this literature in the text.

2. Hsu et al. (2014; ref 12 in this manuscript) reported that a point mutation in the yeast RTR1 gene (Rtr1-E66A) causes a growth defect similar to that of RTR1 deletion (rtr1 Δ). This residue is conserved in RPAP2 (E93) and was found in this study to interact with RPB5 (see Extended Data Figure 3). Can the authors specify if mutation of this residue would affect RPAP2 interaction with Pol II? In relation to this, does the binding of RPAP2 to the Pol II jaw require phosphorylation of the Pol II subunits, e.g. phosphorylation of T111 of RPB5 (situated opposite to E93)? More specifically, could the authors observe any density in their cryo-EM structure that could correspond to phosphorylation of e.g. T111 of RPB5?

We included the following sentence:

Line 68: "Mutation of residue E66 to alanine in yeast Rtr1 (corresponding to E93 in RPAP2) led to a growth defect phenotype similar to Rtr1 deletion [PMID 24951832] demonstrating the importance of this interaction."

We did not observe additional density around RPB5 T111 that might suggest this residue is phosphorylated.

3. It is unclear to me, whether the competition assays address the association of RPAP2-Pol II with TFIIF. Methods and Results of competition assays could be better explained. Could the authors clarify if binding of TFIIF to Pol II would be sufficient for the dissociation of RPAP2 from Pol II (see line 71-73)? More specifically, could Pol II be simultaneously bound by RPAP2 and TFIIF. As it was reported that a significant fraction of free Pol II in the nucleus associates with TFIIF, it should be possible to assess whether a ternary Pol II-TFIIF-RPAP2 can be purified (Rani et al. 2004, PMID: 14749386).

Our PIC assembly protocol involves prior formation of a Pol II-TFIIF complex. This is now clearly indicated in the methods. For technical reasons (protein stabilities in various buffers), we could not form a ternary Pol II-TFIIF-RPAP2 complex. However, the fact that RPAP2 is displaced by nucleic acids in an elongation complex of Pol II in the absence of TFIIF suggests that it is the downstream DNA that is displacing RPAP2 and not TFIIF.

4. Could the authors discuss if the structural differences of the insertion between RPAP2 (β -sheets) and Rtr1 (α -helix) would result into functional differences (stronger/weaker interactions with RPB1 etc.)?

The structural differences between Pol II bound RPAP2 and the crystal structure of yeast Rtr1 could lead to stronger binding of RPAP2 to Pol II compared to Rtr1.

We included the following sentence in the manuscript:

Line 60: The contacts between this metazoan-specific region of RPAP2 and RPB1 may lead to a stronger binding of RPAP2 to Pol II in metazoans compared to binding of Rtr1 to Pol II in yeast.

REVIEWERS' COMMENTS:

Reviewer #1 (Remarks to the Author):

The authors addressed my comments and the manuscript improved. I have no further suggestions.

Reviewer #2 (Remarks to the Author):

The authors have satisfied previous concerns.

I still have a very minor point concerning the competition assays with the core PIC.

In the Methods section, it is said (lines 271-277): "... Pol II, RPAP2(1-215), and TFIIF were mixed...

We then mixed Pol II-RPAP2(1-215)-TFIIF and dsDNA – TFIIA variant – TFIIB – TBP.... We

could not form a stable Pol II-TFIIF-RPAP2 ternary complex."

But in the main text, the authors write: "...we incubated the Pol II-RPAP2(1-215) complex with the transcription initiation factor (TF)-IIF and preformed upstream promoter complex containing promoter DNA, TFIIA, TFIIB and the TATA box-binding protein (TBP)."

I do understand that, as a ternary complex (Pol II-RPAP2-TFIIF) could not be obtained, the authors incubated a Pol II-RPAP2 complex with TFIIF and a preformed upstream promoter complex. If this is correct, the authors could clarify this point by modifying the Methods accordingly.

Response to reviewer comments
Manuscript COMMSBIO-20-3505-T
Fianu et al. "Structure of RNA polymerase II-RPAP2 complex"

Responses are in italics

Reviewer #2 (Remarks to the Author):

The authors have satisfied previous concerns.

I still have a very minor point concerning the competition assays with the core PIC. In the Methods section, it is said (lines 271-277): "... Pol II, RPAP2(1-215), and TFIIF were mixed... We then mixed **Pol II-RPAP2(1-215)-TFIIF and dsDNA – TFIIA variant – TFIIB – TBP**.... We could not form a stable Pol II-TFIIF-RPAP2 ternary complex."

But in the main text, the authors write: "...we incubated the Pol II-RPAP2(1-215) complex with the transcription initiation factor (TF)-IIF and preformed upstream promoter complex containing promoter DNA, TFIIA, TFIIB and the TATA box-binding protein (TBP)."

I do understand that, as a ternary complex (Pol II-RPAP2-TFIIF) could not be obtained, the authors incubated a Pol II-RPAP2 complex with TFIIF and a preformed upstream promoter complex. If this is correct, the authors could clarify this point by modifying the Methods accordingly.

We have clarified this point and now reads:

Line 237: To test binding of RPAP2(1-215) to Pol II upon formation of a core PIC, we first tested the formation of a Pol II-TFIIF-RPAP2 ternary complex. We mixed Pol II, RPAP2(1-215), and TFIIF and incubated for 30 min at room temperature (RT). We could not form a stable Pol II-TFIIF-RPAP2 ternary complex. In parallel, dsDNA (same as above), TFIIA variant, TFIIB and TBP were mixed and incubated for 2 min at RT to form an upstream promoter complex. We then mixed the Pol II-RPAP2(1-215) complex, TFIIF and the upstream promoter complex, diluted to a final volume of 40 μ L in complex buffer and incubated for 30 min at 30 °C.